# Zero-Shot Sand-Dust Image Restoration

**DOI:** 10.3390/s25061889

**Published:** 2025-03-18

**Authors:** Fei Shi, Zhenhong Jia, Yanyun Zhou

**Affiliations:** 1School of Computer Science and Technology, Xinjiang University, Urumqi 830046, China; sigofei@xju.edu.cn (F.S.); zhouyanyun@xju.edu.cn (Y.Z.); 2Key Laboratory of Signal Detection and Processing, Xinjiang University, Urumqi 830046, China

**Keywords:** sand-dust image, zero-shot learning, image restoration, atmospheric scattering physics model

## Abstract

Natural sand-dust weather is complicated, and synthetic sand-dust datasets cannot accurately reflect the properties of real sand-dust images. Sand-dust image enhancement and restoration methods that are based on enhancement, on priors, or on data-driven may not perform well in some scenes. Therefore, it is important to develop a robust sand-dust image restoration method to improve the information processing ability of computer vision. In this paper, we propose a new zero-shot learning method based on an atmospheric scattering physics model to restore sand-dust images. The technique has two advantages: First, as it is unsupervised, the model can be trained without any prior knowledge or image pairs. Second, the method obtains transmission and atmospheric light by learning and inferring from a single real sand-dust image. Extensive experiments are performed and evaluated both qualitatively and quantitatively. The results show that the proposed method works better than the state-of-the-art algorithms for enhancing and restoring sand-dust images.

## 1. Introduction

Images captured in inclement weather conditions, such as snow [1], rain [2], haze [3], underwater [4], and sandstorms, often exhibit poor visual quality. The absorption of light by the turbid media reduces image contrast and obscures the details of the captured images. Particularly in sand-dust weather, the absorption and scattering of wavelength-dependent light by sand-dust particles can introduce color casts in the captured images [5,6]. Images with poor visibility due to sand-dust conditions can significantly degrade the performance of various vision tasks, including object recognition, intelligent transportation systems, surveillance, and oil pipeline inspection. To enhance visual perception, it is essential to develop robust techniques for image enhancement and restoration.

Generally, the degradation of sand-dust images is primarily influenced by light absorption and scattering, as well as by the imaging equipment [7,8,9,10]. Specifically, the degradation of sand-dust images can be attributed to the following factors: (1) Visible light is scattered by the properties of the turbid medium in the environment. Consequently, ambient light comprises both atmospheric light and scattered light. (2) The imaging process relies on the radiation intensity of light from the object. However, the absorption of visible light by the atmospheric medium reduces the reflection intensity. Moreover, the attenuation of light is related to the distance between the target and the imaging equipment. (3) Image quality degradation is also caused by the resource limitations of imaging equipment, such as storage and compression, etc. From the perspective of the imaging process, degraded sand-dust images typically exhibit characteristics such as color casts, low contrast, and blurring.

Many sand-dust image enhancement and restoration algorithms have been proposed and can be broadly categorized into three categories: (1) image enhancement methods, (2) image restoration methods, and (3) deep learning-based approaches. Image enhancement methods encompass color space transformation and enhancement [11,12,13,14], various improved histogram equalization techniques [8,9,10,13,15], white balance and filtering approaches [15,16], singular value decomposition methods [7,17], and tensor least squares optimization methods [18]. Additionally, methods based on the enhanced retinex model [19,20] and those based on diverse image fusion techniques [21,22,23,24] have been proposed. Although these enhancement methods have achieved marked improvements in contrast and tone adjustment, they rely on various hypotheses and priors. The performance of these algorithms is influenced by the varying degrees of degradation in sand-dust images. Additionally, the color casts and sand-dust in the degraded images cannot be adequately removed.

To enhance image details more effectively, numerous image restoration methods based on atmospheric scattering models have been proposed. These methods employ various strategies to optimize transmission and atmospheric light, such as imposing information loss constraints [25], utilizing diverse optimization algorithms (e.g., genetic algorithms) [26,27], and addressing ambient light differences [28] and light compensation [29]. Additionally, to estimate the unconstrained atmospheric model, several improved methods based on the dark channel prior have been developed to enhance the restoration quality of sand-dust images. These methods include blue channel inversion [6], adaptive technologies [30,31], gamma correction techniques [32], Laplacian distribution models [31], regression analysis [5], and the rank-one prior [33]. These methods can accurately restore sand-dust images, but they rely on prior knowledge to calculate the transmission and atmospheric light in the model, which limits the robustness and generalizability of the algorithms.

With the development of deep learning, deep learning-based methods have been tested for processing sand-dust images [34,35,36,37,38,39,40]. However, real sand-dust images are highly complex, and synthetic sand-dust images fail to capture their characteristics accurately. Consequently, methods that rely on synthetic sand-dust image data for model training often exhibit poor performance on real sand-dust images.

To address the aforementioned issues, this paper proposes a zero-shot realistic sand-dust image restoration method. We view the degraded image as a combination of a clear image, transmission map, and atmospheric light. Based on this concept, we design a specific network model to estimate the parameters in the atmospheric scattering physical model. Subsequently, we restore clear images using the inverse model of the atmospheric scattering model. The proposed method is markedly different from existing deep learning-based methods and zero-shot learning methods, primarily in the following aspects: First, unlike data-driven deep learning methods, the proposed method employs an unsupervised and zero-shot learning approach, requiring only a degraded image as input to produce a corresponding clear image. Second, compared with existing zero-shot learning methods, the proposed method relies entirely on the atmospheric scattering model for network design and does not require any prior knowledge or controlled parameters for model training.

To highlight them, this study makes the following primary contributions:To the best of our knowledge, this paper is the first to propose a zero-shot learning method for sand-dust image restoration. Despite being a zero-shot learning method, the proposed method achieves superior performance in restoring real sand-dust images.A new joint-learning network structure model is proposed, and the network model is designed entirely based on the imaging principles of the physical model of atmospheric scattering. A modified U-Net network is used to obtain a clear image and transmission map. Considering the relationship among the input image, clear image and transmission map in the atmospheric scattering model, a convolutional network and fully connected layers are designed to obtain atmospheric light.

The rest of this paper is organized as follows. Section 2 details various methods for studying sand-dust images. Section 3 presents the method proposed in this paper. Section 4 presents the experimental results. Section 5 summarizes the approach proposed in this paper.

## 2. Related Work

### 2.1. Sand-Dust Image Enhancement and Restoration Methods

In recent years, an increasing number of scholars have conducted extensive research and made significant progress in image enhancement and restoration. While many scholars have focused on recovering images captured under specific weather conditions such as rain, snow, and fog, the enhancement and restoration of sand-dust images have also emerged as a hot topic in recent years. Several traditional image enhancement methods have been employed to improve the quality of sand-dust images [7,8,9,10,11,12,13,14,15,16,18,41,42]. For example, Shi [9] proposed an adaptive histogram equalization method based on the normalized gamma transform. This method only enhanced the visual quality but did not remove dust haze from sand-dust images, and it fails in extreme cases. Cheng [16] proposed a method involving blue channel compensation and guided filtering to enhance sand-dust images. Since the blue channel of sand-dust images decays the fastest, blue channel compensation was initially applied. The white balance technique was then used to adjust the color cast, and finally, guided filtering was employed to enhance contrast and detail. However, this method relies on the quality of the guided image and the compensation coefficient. Park [10] proposed a continuous color correction method based on coincident histograms. First, the color correction method relies on mean and variance statistics for color correction. Then, the image is normalized using the green channel mean preservation method. Finally, the maximum histogram overlap method is employed to remove color casts. However, this algorithm depends on the number of channel histogram offset steps. Xu [18] discovered that a key intrinsic characteristic of sandstorm-free outdoor scenes is the rough similarity in the contours of the RGB channels. Based on this finding, Xu proposed a tensor least squares optimization method to enhance sand-dust images. This method improves contrast and reduces chromatic aberration to some extent. However, it also amplifies image noise during the enhancement process. To enhance the details of the image, some researchers introduced fusion technology [21,22,23,24]. For example, Fu [21] proposed an approach that employs an adaptive statistics strategy to correct color. Subsequently, two gamma coefficients are applied to enhance the image. Finally, a weight fusion strategy is used to fuse the image. However, this method relies on manual parameters. Although the aforementioned methods have achieved satisfactory results in enhancing sand-dust images, their restoration performance is contingent upon the settings of certain hyperparameters.

To restore image details while enhancing the image, methods based on the atmospheric scattering physical model and retinex theory have been employed to restore and enhance sand-dust images [5,6,17,25,26,27,28,29,30,31,32,33,43,44,45]. These methods, which leverage atmospheric physical models, have achieved marked progress. In particular, dark channel prior (DCP) methods, which are based on atmospheric scattering models, have achieved significant results in image dehazing. These methods have been adapted and improved for processing sand-dust images. For example, Peng [5] conducted regression analysis on the scene depth of the input image to obtain depth information and calculated the ambient light difference to estimate the transmission map. Finally, adaptive gamma correction was employed to eliminate color casts. However, this method relies on scene depth estimation and fails when the sand image has lights at different depths. Shi [43] proposed a sand-dust image enhancement method based on the halo-reduced dark channel prior. This method includes color correction based on the gray world theory, sand-dust removal, and gamma correction. Gao [6] proposed a method involving the reversal of the blue channel prior for restoring sand-dust images. This method relies on adjusting the blue channel, which may introduce a new blue color cast in the restored image. Liu [33] introduce a rank-one matrix prior to estimate scattering map, but this method can not completely remove sand-dust in the image. In addition, the method based on retinex theory was also applied to enhance the dust image [19,20]. For example, Li [19] proposed an improved retinex method, which focused on a noise map and introduced the L1 regularization term to optimize brightness and reflectivity. White balance and gamma correction technology were used to enhance contrast in [20] by Kenk, and the Laplacian pyramid filter was used to enhance the details of reflected components. Although model-based methods have achieved significant progress, their effectiveness depends on the prior assumptions and the distribution of the real data. If the prior knowledge is not applicable to certain scenes, the desired results may not be achieved.

Currently, many deep learning-based methods have achieved significant results in dehazing, rain removal, and snow removal. However, due to the lack of large-scale paired real degraded image datasets, most learning-based methods rely on synthetic degraded images to train network models.Because paired real sand-dust image data are difficult to obtain, Si [34] proposed a synthesis algorithm for sand-dust images, which attempts to recover sand-dust images using a deep learning approach. To obtain better results, Si et al. [35] proposed a fusion algorithm combining a color correction algorithm with deep learning. Shi [36] proposed a convolutional neural network incorporating color recovery to enhance sand-dust images. Gao [38] proposed a novel two-in-one low-visibility enhancement network that can enhance both haze and sand-dust images. However, it is challenging to achieve satisfactory results on real sand-dust images. As reported by Golts [46], synthetic data cannot describe the characteristics of real degraded images, leading to a domain shift. To reduce the impact of synthetic data on the model, Ding [37] proposed a single-image sand-dust restoration algorithm based on style transformation and unsupervised adversarial learning. Gao [39] proposed a two-step unsupervised approach for enhancing sand-dust images. First, an adaptive color cast correction factor was developed. Then, an unsupervised generative adversarial network (GAN) was used to further enhance the images. However, this method may introduce color distortion and halos in some sand-dust scenes. Meng [40] proposed an unpaired GAN method for image dedusting based on retinex. However, this method fails in high-concentration and complex scenes. Despite their effectiveness in dedusting, deep learning-based methods still rely on artificial priors and synthetic data, which can limit the quality of image restoration. To address the lack of paired real sand-dust samples and reduce the dependence on prior knowledge, inspired by atmospheric scattering models and zero-shot learning, we propose a zero-shot sand-dust image enhancement method.

### 2.2. U-Net Architecture

Olaf [47] initially proposed the U-Net architecture for biomedical image segmentation, utilizing an encoder–decoder framework augmented with skip connections to fuse shallow detail with deep semantic information, thereby achieving remarkable performance even with limited sample data. Some improved U-Net network frameworks have been proposed for various scenarios. Zhou [48] proposed an advanced medical image segmentation architecture, termed UNet++, which introduces dense convolutional blocks between the encoder and decoder to gradually fuse multi-level features, and effectively address the issue of detail loss in complex backgrounds. Chen [49] combined the global context modeling capability of Transformer with the local detail capturing ability of U-Net, and proposed the first TransUNet for medical image segmentation, achieving excellent results in the segmentation of small organs and complex boundaries. Cao [50] constructed a U-shaped network based on Swin Transformer blocks, named Swin-Unet, which significantly improved the accuracy of medical image segmentation and edge prediction effects. Liao [51] proposed a distortion rectification generative adversarial network, and the architecture of the generator is designed based on U-Net. This method can directly learn the end-to-end mapping from distorted images to undistorted images, and has achieved good performance on both synthetic and real distorted images. Li [52] proposed a deep learning framework based on residual blocks for solving multiple types of distortions with a single model; it includes the single-model and multi-model distortion estimation networks. This method combines the displacement field prediction of a convolutional neural network with model fitting optimization to achieve general correction for multiple types of distortions. Liao [53] introduced an innovative U-shaped MOWA model, by decoupling motion estimation at the region level and the pixel level, as well as conducting dynamic task-aware prompt learning; it succeeds in addressing multiple image deformation tasks inside a single model. However, when confronted with complex boundary scenarios, the control point prediction of this method may fail. Our network is designed based on the atmospheric scattering model with a U-Net architecture and achieves more accurate results on sand-dust images.

## 3. Proposed Zero-Shot Method

In this section, we introduce the proposed image restoration framework based on the atmospheric scattering model (see Figure 1). This framework includes the adaptive sand-dust image color cast balance method, which is a zero-shot neural network learning model that combines sharp images, transmission, and atmospheric light, and the image restoration process. The following sections provide a detailed introduction.

Above, I(x) represents the input image, $I_{1}\left(x\right)$ indicates the result after color balance algorithm processing, $M_{T}$ and $M_{A}$ denote the atmospheric light and transmission map estimation networks, respectively, and $J_{R}\left(x\right)$ stands for the restored image.

### 3.1. Image Color Balance

Many methods have been used to correct the color casts of sand-dust images, such as the grayscale world hypothesis theory [6,32,43], strategies for adaptive statistics [21], white balance [15,16], color channel compensation [14,23], and histogram equalization [8,10,13]. Because the sand-dust images captured under different weather conditions have different color casts, it is particularly difficult to design an effective method that can adaptively adjust various color casts. In this paper, we propose a simple and effective adaptive color cast adjustment algorithm based on wavelength-dependent light attenuation characteristics. First, Equation (Equation 1) is used for color fine-tuning; this step is particularly important for the images obtained in strong or extremely strong sandstorm environments. Then, the color casts are removed by Equation (Equation 3).(1)$$I_{0}^{c}\left(x\right)=I^{c}\left(x\right)+\Delta^{c}\times I^{g}\left(x\right),$$
where $I_{0}^{c}\left(x\right)$ represents the image after color cast removal, $I^{c}\left(x\right)$ represents the input color image pixel value, *c* is the color channel, $I^{g}\left(x\right)$ denotes the green channel pixel values of a color image, and $\Delta^{c}$ is the weight of the reserved green channel pixel value, which is expressed as Equation (Equation 2):(2)$$\Delta^{c}=\left(m\left(I^{g}\left(x\right)\right)-m\left(I^{c}\left(x\right)\right)\right)\times \left(1-\frac{m(I^{c}\left(x\right))}{avg(m^{c})}\times I^{c}\left(x\right)\right),$$
where $m(I^{c}\left(x\right))$ is the mean value of the corresponding channels of the color image, *c* is the color channel, *g* is the green channel, and $avg(m^{c})$ is the mean value of the three channels of the RGB color image.

To adjust various color deviations better, according to the statistical characteristics that blue channels decay the fastest and red channels decay the slowest in sand-dust images, a method of maximum green mean value preservation, with red channel and reverse green blue channel compensation is proposed in this paper, which is expressed as Equation (Equation 3):(3)$$I_{1}^{c}\left(x\right)=\frac{I_{0}^{c}\left(x\right)-m\left(I_{0}^{c}\left(x\right)\right)}{\max\left(I_{0}^{c}\left(x\right)\right)-\min\left(I_{0}^{c}\left(x\right)\right)}+I_{cb}\left(x\right),$$
where $I_{0}^{c}\left(x\right)$ represents the image after color cast correction, and “max” and “min” represent the operations of finding the maximum value and the minimum value, respectively. $I_{cb}\left(x\right)=\frac{I_{cb1}\left(x\right)+I_{cb2}\left(x\right)+I_{cb3}\left(x\right)}{2}$ is the RGB channel compensation factor, which consists of the following three items. $I_{cb1}\left(x\right)=\max\left(m\left(I_{0}^{g}\left(x\right)\right),1-m\left(I_{0}^{g}\left(x\right)\right)\right)$ is the maximum green mean preservation compensation coefficient, $I_{cb2}\left(x\right)=m\left(I_{0}^{r}\left(x\right)\right)\times \left(1-m\left(I_{0}^{b}\left(x\right)\right)\right)$ is the compensation coefficient of the red channel’s mean value and the reversed blue channel’s mean value, and $I_{cb3}\left(x\right)=m\left(I_{0}^{g}\left(x\right)\right)\times \left(1-m\left(I_{0}^{g}\left(x\right)\right)\right)$ is the compensation coefficient of the inverted green channel’s mean value. Figure 2 shows the color correction results using the color balance algorithm that is proposed in this paper.

### 3.2. Proposed Method Framework

The atmospheric scattering model [54] is commonly used to describe the degradation process of images, which is expressed as(4)I(x)=J(x)×t(x)+A×(1−t(x)),
where *x* is the position of pixels in the image, I(x) indicates the captured degraded image, J(x) represents a sharp image, *A* represents atmospheric light, and t(x) represents medium transmission. The transmission can be expressed as $t\left(x\right)=e^{-\beta d(x)}$, where d(x) is the scene depth between the camera and the object, and β is the atmospheric scattering coefficient. By inverting Equation (Equation 4), the sharp image J(x) can be obtained according to Equation (Equation 5),(5)$$J\left(x\right)=\frac{I(x)-A\times (1-t(x))}{t(x)}.$$

Equation (Equation 4) shows that the degraded image I(x) is formed by the interaction of the sharp image J(x), the transmission map t(x), and the atmospheric light *A*. Therefore, according to the principle of imaging in the atmosphere, a depth network relying on J(x), t(x), and *A* interrelated can be designed to train the model using a single sand-dust image to obtain the transmission map t(x) and the atmospheric light *A*, and finally, a sharp image J(x) can be recovered by Equation (Equation 5). Although the network for generating sharp images J(x) is designed in this model, it is a challenging task to generate high-quality clear images due to the severe degradation of input images. Therefore, the clear image J(x) in the proposed method is only used to constrain the model training. In addition, unlike other degraded images such as fog, sand-dust images exhibit varying degrees of color cast due to the absorption and scattering of light by sand-dust particles. To mitigate the impact of color cast on model training, an effective color cast correction algorithm is proposed to preprocess the input sand-dust images.

Inspired by atmospheric scattering model, a zero-shot learning method for sand-dust image restoration is proposed in this paper. The framework is shown in Figure 3, where Figure 3a shows the zero-shot training model designed based on the atmospheric scattering model, based on the $I_{1}\left(x\right)$ obtained by the color balance algorithm, the transmission map t(x), and the atmospheric light *A*, and sharp image J(x) obtained by the training model $M_{T}$, $M_{A}$, and $M_{J}$. To be more specific, start with a color-corrected input sand-dust image $I_{1}\left(x\right)$. t(x), J(x), and *A* are first estimated by the designed feature-sharing network. Then, the reconstructed image $I_{2}\left(x\right)$ is generated by the atmospheric scattering physical model, and the reconstructed image is used as input to train the model by constraining the structure of atmospheric light $A_{1}$ and $A_{2}$, transmission maps $t_{1}\left(x\right)$ and $t_{2}\left(x\right)$, reconstructed image $I_{2}\left(x\right)$, and input image $I_{1}\left(x\right)$ to obtain *A* and t(x) accurately. The image restoration model is shown in Figure 3b; a sharp image is restored according to Equation (Equation 5).

### 3.3. Zero-Shot Network Architecture

In this section, we introduce the architecture and implement the model. Figure 3a shows the zero-shot learning model framework that is proposed in this paper. Because the model is designed entirely based on the atmospheric scattering physical model, we consider that the three networks $M_{T}$, $M_{J}$, and $M_{A}$ are interrelated. Inspired by U-Net [47], we proposed a new model architecture shown in Figure 4, where the $M_{T}$, $M_{J}$, and $M_{A}$ models share weights. The transmission map estimation network $M_{T}$ located on the left side of the entire network, while the sharp image estimation network $M_{J}$ is another U-shaped network situated on the right side. The atmospheric light estimation network $M_{A}$ is designed based on Equation (Equation 4); it carries out a joint estimation by integrating the outputs of networks $M_{T}$ and $M_{J}$ together with the input image. FCM64-3 indicates converting a 64-channel image into a 3-channel image through fully connected layers. * represents convolution operation.

### 3.4. Loss Function

The idea comes from the widely used atmospheric scattering physical model that describes the image degradation process, and the model is expressed as Equation (Equation 4). Without real clear images and additional information, to better estimate J(x), t(x), and *A* depending on the feature information of a single image, we define Equation (Equation 6) as the total loss function to jointly train the neural network model:(6)$$L=\omega_{0}L_{V}+\omega_{1}L_{A}+\omega_{2}L_{T}+\omega_{3}L_{S},$$
where $L_{V}$ is the gradient loss of the reconstructed clear image J(x) and the model-generated clear image $J_{1}\left(x\right)$; $L_{A}$ is defined as the atmospheric light loss between $A_{1}$ and $A_{2}$; $L_{T}$ is the transmission loss of the sharp image generated by the model; and $L_{S}$ is the image saturation-penalized loss function.

Specifically, $L_{V}$ ensures neighborhood consistency in the restored image and thus reduces the local variation. The loss function is defined as(7)$$L_{V}=\sum_{x,c,k}\left({\left|\nabla_{H}J_{k}^{c}\left(x\right)\right|}^{2}+{\left|\nabla_{V}J_{k}^{c}\left(x\right)\right|}^{2}\right),$$
where *x* is the position of pixels in the image, *c* represents the color channel of the color image, k∈(0,1) denotes the reconstructed clear image $J_{0}\left(x\right)$ and the model-generated clear image $J_{1}\left(x\right)$, respectively, and $\nabla_{H}$ and $\nabla_{V}$ represent the horizontal and vertical gradient operators, respectively.

The transmission loss $L_{T}$ and the atmospheric light loss $L_{A}$ are two important loss functions, which primarily keep $t_{1}\left(x\right)$ equal to $t_{2}\left(x\right)$ and $A_{1}$ equal to $A_{2}$. The transmission loss and atmospheric light loss are defined as Equations (Equation 8) and (Equation 9), respectively.(8)$$L_{T}=\sum_{x}{∥t_{1}\left(x\right)-t_{2}\left(x\right)∥}^{2},$$(9)$$L_{A}={∥A_{1}-A_{2}∥}^{2},$$
where *x* represents image pixels; $t_{1}\left(x\right)=M_{T}\left(I_{1}\left(x\right)\right)$, $t_{2}\left(x\right)=M_{T}\left(I_{2}\left(x\right)\right)$, $A_{1}=M_{A}\left(I_{1}\left(x\right)\right)$, and $A_{2}=M_{A}\left(I_{2}\left(x\right)\right)$; and $M_{A}$ and $M_{T}$ are the atmospheric light and transmission estimation networks proposed in this paper. We also included the saturation penalty loss $L_{S}$ in [55].

## 4. Experiments

To verify the effectiveness of the proposed algorithm, we conducted experiments using numerous images collected under various sand-dust weather conditions and compared the results with those of 12 state-of-the-art sand-dust image enhancement and restoration algorithms using six metrics. The following sections will introduce the experimental configuration and the qualitative and quantitative evaluation results.

### 4.1. Experimental Configurations

In this section, we describe the dataset, comparison methods, evaluation metrics, and training procedure used in the experiments.

(1) Dataset: Due to the lack of a unified public dataset for sand-dust images, we collected 1070 real images under various sand-dust weather conditions from the Internet to more thoroughly validate the algorithm. Additionally, we captured 460 high-resolution real images under sand-dust weather conditions in Xinjiang for this study. All images were cropped to a size of 640 × 480.

(2) Comparative Methods: For comprehensive comparison, the proposed method is compared with 12 state-of-the-art sand-dust image processing algorithms, including the generalized dark channel prior method (GDCP) [5], reverse blue channel prior method (RBCP) [6], color balance method based on maximum histogram overlap (SCBCH) [10], adaptive histogram equalization method based on normalized gamma transform (NGT) [9], rank-one matrix prior (ROP) [33], blue channel compensation and guided filtering technique (BCGF) [16], tensor least squares optimization method (TLS) [18], image fusion-based approach (FBE) [21], two-in-one low-visibility enhancement network (TOENet) [38], dark channel prior method based on halo elimination (HDCP) [43], unpaired GANs method for image dedusting (DedustGAN) [40], and unsupervised generative adversarial network for sand-dust image enhancement (SIENet) [39].

(3) Evaluation Metrics: Because there are no corresponding sharp images, non-reference evaluation methods are applied to assess the performance of the algorithms. Six evaluation metrics are introduced for comprehensive assessment, including the evaluation of image integrity and authenticity based on distortion identification (DIIVINE) [56], blind image evaluation method based on scene statistics and perceptual features (NPQI) [57], natural scene image evaluation method based on multivariate Gaussian model (NIQE) [58] and the visual edge restoration percentage *e*, black and white pixel saturation percentage σ, and contrast recovery percentage $\bar{r}$ as suggested in [59]. The smaller the DIIVINE the lower the distortion of the recovered image and the higher the quality of the image; the smaller the NPQI and NIQE, the better the quality of the recovered image; the larger the *e* and the closer the σ is to 0, the better the image quality; and the larger the $\bar{r}$, the higher the contrast of the recovered image and the better the quality of the image.

(4) Training Procedure: The loss function is shown in Equation (Equation 6), where the coefficients are set as follows: $\omega_{0}=\omega_{3}=0.001$, $\omega_{1}=\omega_{2}=1$. Rotation and mirroring improve the input image in the training stage. This method has been shown to be useful in unsupervised learning [55,60]. The weights in the model follow a normal distribution with a mean value of 0 and a standard deviation of 0.001. The optimization process uses the default Adam optimizer [61], and the learning rate is set to 0.001. The model is trained 1000 times on an 3090Ti GPU from NVIDIA, Santa Clara, CA, USA. All images used in the experiment were cropped to 640 × 480.

### 4.2. Qualitative Evaluation

To more thoroughly verify the algorithm performance, we collected sand-dust images captured under five types of sand-dust weather conditions: floating dust, sand, sandstorms, strong sandstorms, and extremely strong sandstorms. Extensive comparison experiments were conducted using these sand-dust images, and the experimental results are shown in Figure 5, Figure 6, Figure 7, Figure 8 and Figure 9.

Figure 5 shows the experimental results of various algorithms for images captured under sand-dust weather conditions in Xinjiang. The images obtained in this scenario are more similar to haze images. This similarity is due to the small size of sand-dust particles in the air, which results in minimal absorption and scattering of light and images with almost no color casts and only a thin dust mist. Figure 5 shows that all algorithms achieve some results in processing floating dust images. However, methods GDCP [5], TLS [18], ROP [33], TOENet [38], and HDCP [43] cannot remove dust haze from some images, such as the fourth image. Color distortion appears in images processed by RBCP [6] and SIENet [39]. The images processed with methods SCBCH [10], NGT [9], BCGF [16], FBE [21], and ours obtained better visual effects, and the contrast of the images was improved. The image restored with the DedustGAN [40] is distorted, which is evident in the sky region in the fifth image.

Figure 6 shows the processing results of various methods for sand-dust images with weak color casts captured under sand-dust weather in Xinjiang. As shown in Figure 6, FBE [21] can obtain better results with more natural image tones. GDCP [5] and TOENet [38] cannot remove color cast. The images recovered with RBCP [6], BCGF [16], and DedustGAN [40] show local darkness and color distortions. The SCBCH [10], ROP [33], NGT [9], and TLS [18] methods cannot remove the dust haze from the image, but improve the contrast of the restored image. Images restored by SIENet [39] and HDCP [43] have enhanced detail and noise, but the restored image is distorted. Due to the influence of prior knowledge and synthetic dataset training models, some algorithms obtain poor results when processing sand-dust images with slight color casts, mainly because the estimated parameters are inaccurate.

The results of images captured under sandstorm weather are shown in Figure 7. The SCBCH [10] and FBE [21] methods can eliminate color casts more effectively than other methods. The GDCP [5] and TOENet [38] methods cannot eliminate color casts. The images restored by other comparison algorithms such as SIENet [39] and RBCP [6] have varying degrees of color casts. The NGT [9], DedustGAN [40], and SIENet [39] methods failed to effectively restore images, as they were unable to adequately correct for image distortion.

Figure 8 and Figure 9 illustrate the results of images captured under severe and extremely strong sandstorm conditions, respectively. As shown in Figure 8 and Figure 9, although the above algorithms achieve satisfactory results for floating dust images and sand-dust images with weak color casts, they fail to effectively recover images taken under strong dust storm conditions with significant color shifts. Methods SCBCH [10] and SIENet [39] can effectively eliminate color casts but do not enhance the image. Other comparison algorithms such as GDCP [5], RBCP [6], NGT [9], ROP [33], BCGF [16], TLS [18], FBE [21], TOENet [38], HDCP [43], and DedustGAN [40] fail to produce satisfactory results for images captured under strong sandstorm conditions, and not only fail to eliminate color deviation, but also cause new color deviations as well as distortions in the recovered image. In contrast, the proposed algorithm consistently yields superior results across various sand-dust scenarios.

In general, the subjective comparison results of various sand-dust images demonstrate that the proposed algorithm outperforms the state-of-the-art methods. The proposed algorithm effectively restores images captured in diverse sand-dust scenarios, achieving superior visual effects. Specifically, the contrast of the restored images is enhanced, and the details of objects within the images are more clearly delineated.

### 4.3. Quantitative Evaluation

To comprehensively evaluate the proposed algorithm, we employed a suite of quantitative metrics, including *e*, σ, $\bar{r}$, DIIVINE, NPQI, and NIQE, as detailed in Section 4.1. These metrics were used to compare the performance of the proposed algorithm against 12 state-of-the-art comparison algorithms.

Table 1 shows the average evaluation results of the 25 images in Figure 5, Figure 6, Figure 7, Figure 8 and Figure 9. As shown in Table 1, the proposed algorithm can obtain larger *e* and $\bar{r}$ values as well as a smaller smaller σ, Additionally, it consistently ranks among the top-performing algorithms in other metrics. The lowest NIQE and NPQI values obtained by the proposed algorithm suggest that the restored images exhibit superior quality and are rich in natural scene statistical feature information. Although the DIIVINE metric for RBCP, SCBCH, and TLS consistently outperform the proposed method, the subjective results are unsatisfactory, with the restored images exhibiting low brightness and the presence of sand and dust fog. The high DIIVINE scores may be attributed to the feature extraction based on statistical models and the distortion classification mechanism, which fail to fully encompass the diversity of complex visual perception. Haze and low lighting conditions can potentially “deceive” the algorithm, causing its statistical features to align more closely with the high-score patterns in the training data.

To comprehensively evaluate the performance of the proposed algorithm, we collected a large number of real sand-dust images. Table 2 presents the average results of 1070 sand-dust images with various color casts collected from the Internet. Additionally, Table 3 provides the average evaluation results of 460 images captured under real sand-dust weather conditions. The *e* in Table 2 cannot be calculated due to the high distortion of the compressed sand-dust image collected from the Internet. Although the HDCP [43] method achieves the best $\bar{r}$, the real restoration performance is poor, as shown in Figure 8 and Figure 9. In addition, the proposed algorithm achieved the best NIQE and NPQI scores and superior DIIVINE values, indicating its ability to produce high-quality restored images with minimal distortion and rich natural scene statistical feature information when processing various distorted sand-dust images.

To mitigate the impact of image distortion on the performance evaluation of the algorithm, we collected 460 high-resolution sand-dust images from real-world scenarios. These images were used to comprehensively measure the algorithm’s performance. The results in Table 3 also show that the overall performance of the proposed method is better than other state-of-the-art algorithms. The qualitative analyses in Figure 5, Figure 6, Figure 7, Figure 8 and Figure 9 and quantitative analyses in Table 1, Table 2 and Table 3 show that the sand-dust images recovered by the proposed algorithm have better visual qualities, and the performance of the proposed algorithm is better. This is due to the fact that the designed U-Net network structure extracts features better.

### 4.4. Ablation Study

To verify the effectiveness of the loss functions used for model training, ablation experiments are conducted for different loss functions in this section. Figure 10 shows the ablation results of the loss function. As shown in Figure 10c, when the atmospheric light loss function is not used, the restored image is darker overall. As shown in Figure 10d, when the transmission loss function is not used, the fog in the recovered image cannot be completely eliminated. Figure 10e,f show that without a saturation penalty function for bright and dark pixels, the reconstructed image is either very brilliant or very dark. As shown in Figure 10g, the main contribution of the gradient loss function is to improve the fidelity of the recovered image.

To further evaluate the performance of the proposed algorithm, we conducted a quantitative assessment of the loss function using 1530 real sand-dust images. The quantitative results are shown in Table 4. As can be seen from Table 4, the joint effect of all loss functions achieved significant improvements.

## 5. Conclusions

In this paper, we propose a zero-shot learning-based dust image restoration method based on the imaging model of images in the atmosphere that does not rely on prior knowledge of images or the setting of hyperparameters. The training of the model can be completed by inputting only a single sand-dust image, and the sharp image can be recovered by the atmospheric scattering model. We conduct experiments on many real sand-dust images and perform qualitative and quantitative analyses. Experimental results show that the proposed algorithm can restore images under various sand-dust weather conditions. The proposed method can eliminate color casts and enhance contrast and details of the images more effectively than existing algorithms, and the robustness and performance of the proposed method are better than those of state-of-the-art algorithms. In the algorithmic framework proposed in this paper, color cast elimination is considered during preprocessing for sand-dust images before model training. In the future, we plan to investigate the design of a model that can eliminate color casts adaptively and restore sand-dust images accurately.

## Figures and Tables

**Figure 1 sensors-25-01889-f001:**
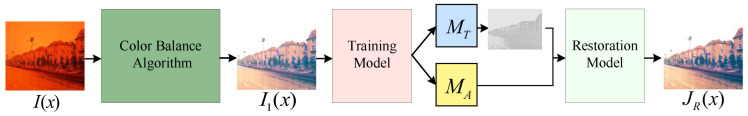
Proposed image restoration framework based on the atmospheric scattering model.

**Figure 2 sensors-25-01889-f002:**
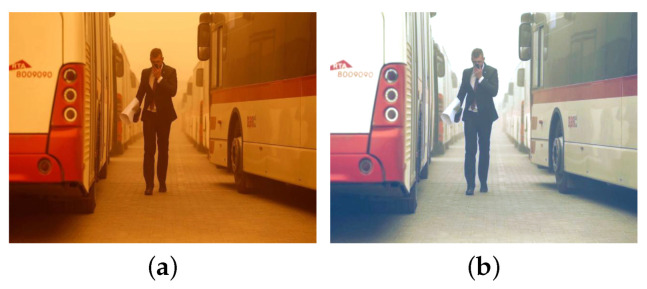
Results of the proposed color balance method. (**a**) Sand-dust image. (**b**) Color balance result by proposed method.

**Figure 3 sensors-25-01889-f003:**
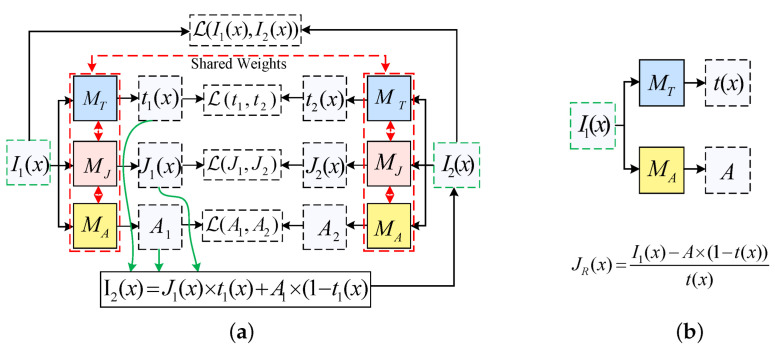
Proposed zero-shot learning framework for sand-dust images restoration. (**a**) Training model. (**b**) Restoration model.

**Figure 4 sensors-25-01889-f004:**
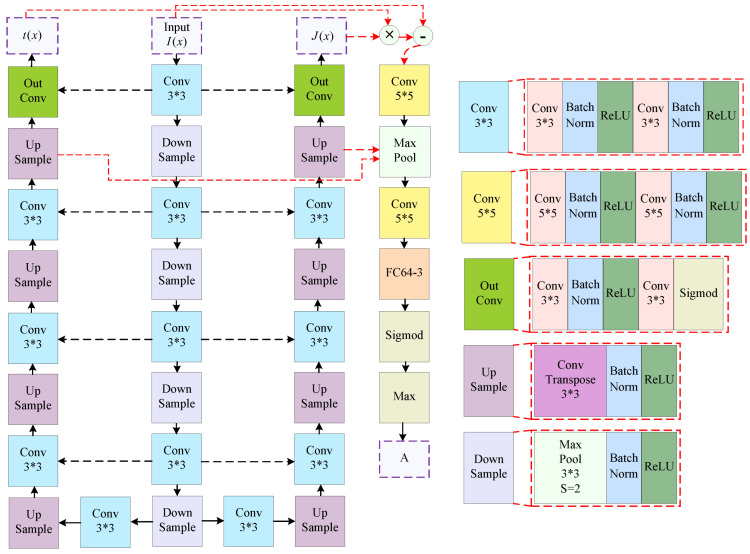
Architecture of our proposed method.

**Figure 5 sensors-25-01889-f005:**
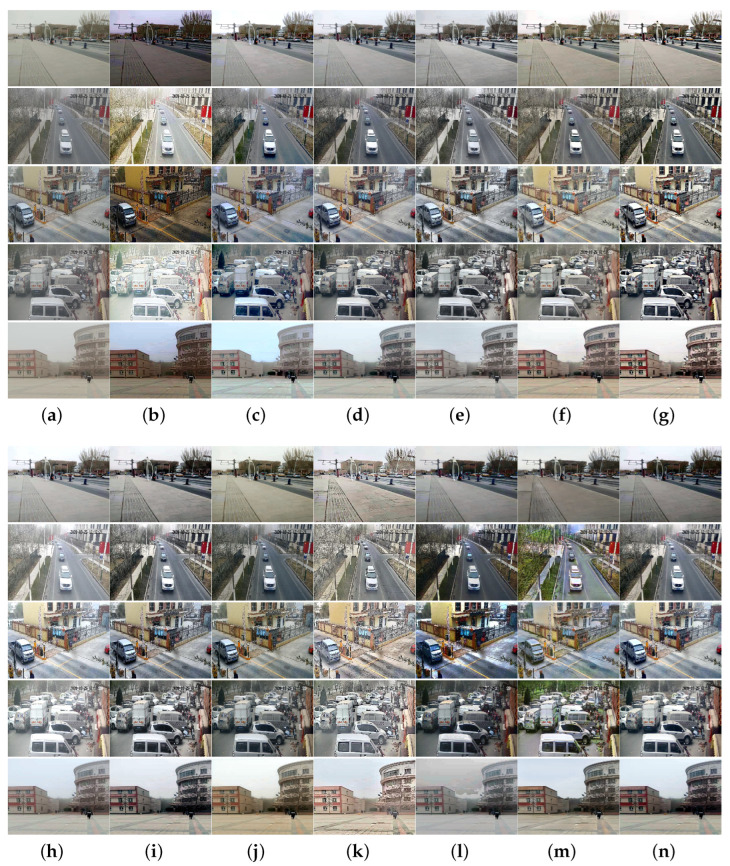
Visual comparisons in real floating dust scenes. (**a**) Sand-dust images. (**b**) GDCP [5]. (**c**) RBCP [6]. (**d**) SCBCH [10]. (**e**) NGT [9]. (**f**) ROP [33]. (**g**) BCGF [16]. (**h**) TLS [18]. (**i**) FBE [21]. (**j**) TOENet [38]. (**k**) HDCP [43]. (**l**) DedustGAN [40]. (**m**) SIENet [39]. (**n**) Ours.

**Figure 6 sensors-25-01889-f006:**
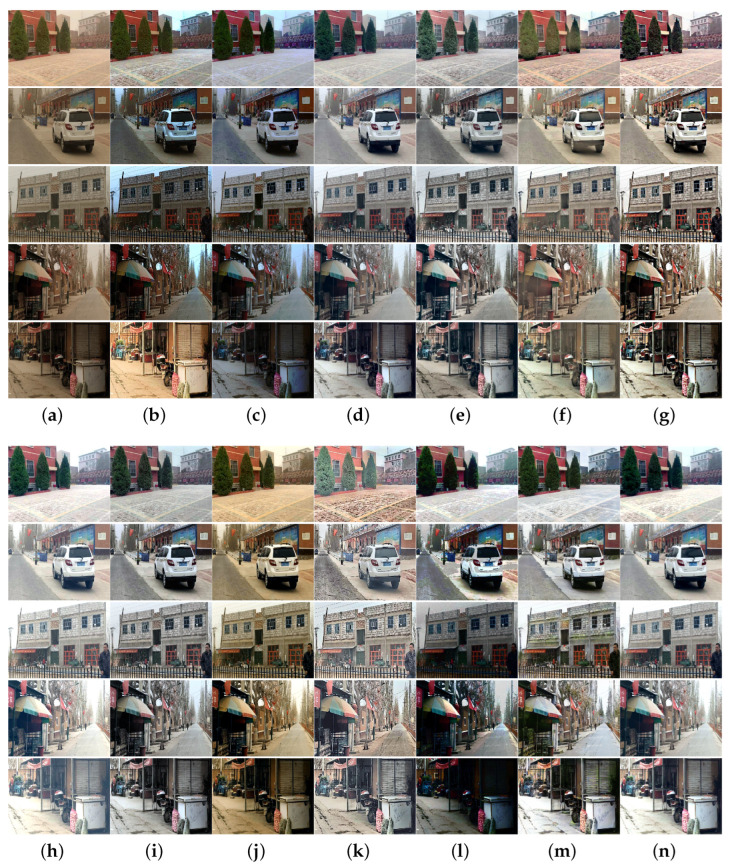
Visual comparisons of real sand-dust images with color cast. (**a**) Sand-dust images. (**b**) GDCP [5]. (**c**) RBCP [6]. (**d**) SCBCH [10]. (**e**) NGT [9]. (**f**) ROP [33]. (**g**) BCGF [16]. (**h**) TLS [18]. (**i**) FBE [21]. (**j**) TOENet [38]. (**k**) HDCP [43]. (**l**) DedustGAN [40]. (**m**) SIENet [39]. (**n**) Ours.

**Figure 7 sensors-25-01889-f007:**
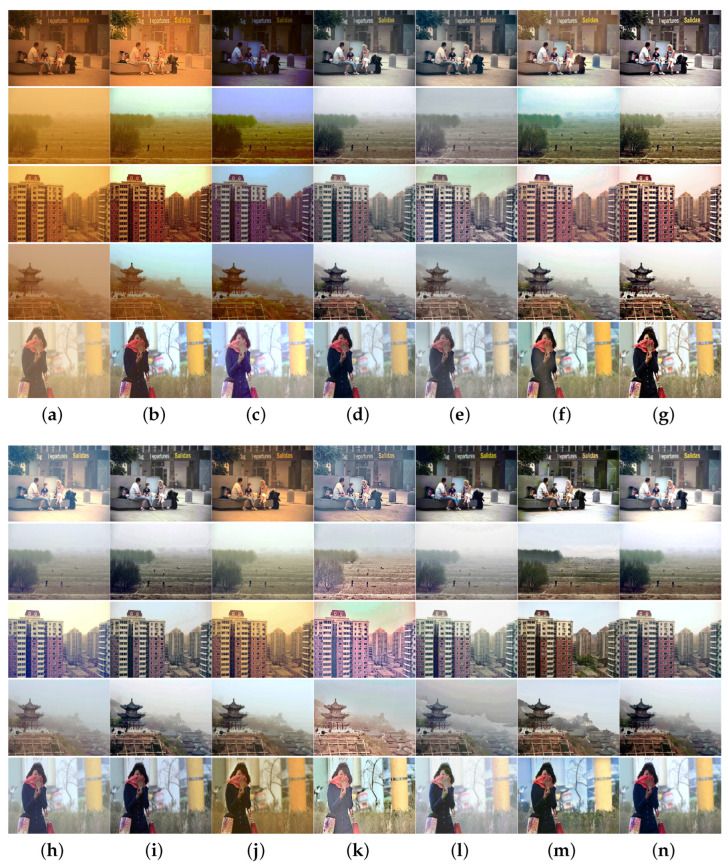
Visual comparisons in sandstorm scenes. (**a**) Sand-dust images. (**b**) GDCP [5]. (**c**) RBCP [6]. (**d**) SCBCH [10]. (**e**) NGT [9]. (**f**) ROP [33]. (**g**) BCGF [16]. (**h**) TLS [18]. (**i**) FBE [21]. (**j**) TOENet [38]. (**k**) HDCP [43]. (**l**) DedustGAN [40]. (**m**) SIENet [39]. (**n**) Ours.

**Figure 8 sensors-25-01889-f008:**
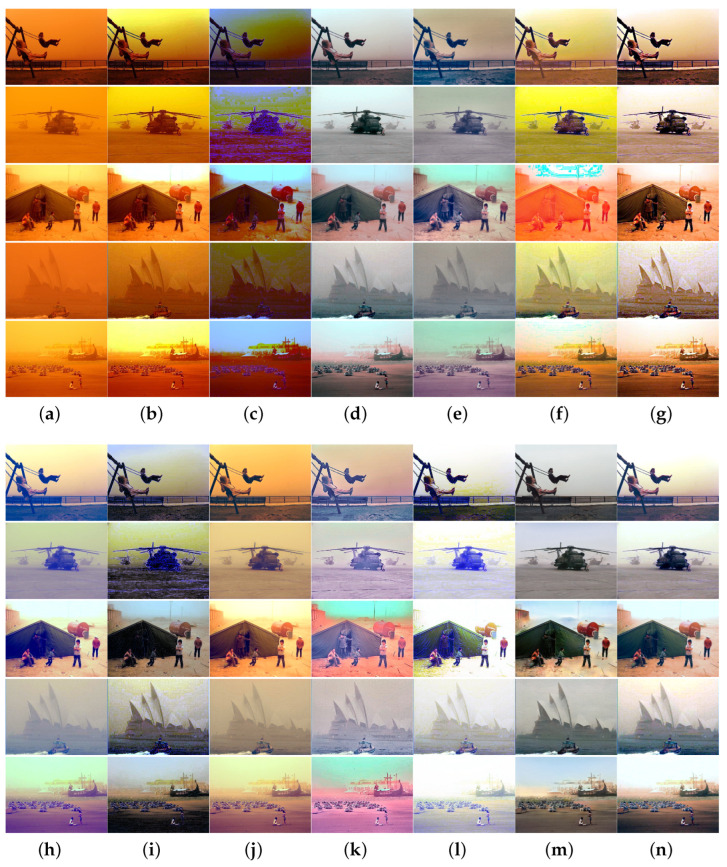
Visual comparisons in strong sandstorm scenes. (**a**) Sand-dust images. (**b**) GDCP [5]. (**c**) RBCP [6]. (**d**) SCBCH [10]. (**e**) NGT [9]. (**f**) ROP [33]. (**g**) BCGF [16]. (**h**) TLS [18]. (**i**) FBE [21]. (**j**) TOENet [38]. (**k**) HDCP [43]. (**l**) DedustGAN [40]. (**m**) SIENet [39]. (**n**) Ours.

**Figure 9 sensors-25-01889-f009:**
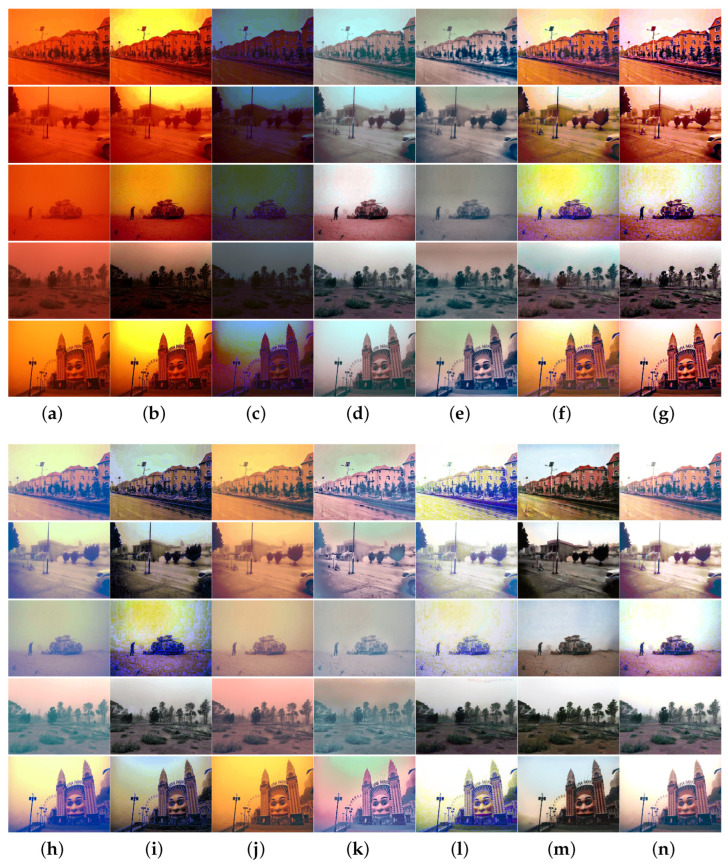
Visual comparisons in extremely strong sandstorm scenes. (**a**) Sand-dust images. (**b**) GDCP [5]. (**c**) RBCP [6]. (**d**) SCBCH [10]. (**e**) NGT [9]. (**f**) ROP [33]. (**g**) BCGF [16]. (**h**) TLS [18]. (**i**) FBE [21]. (**j**) TOENet [38]. (**k**) HDCP [43]. (**l**) DedustGAN [40]. (**m**) SIENet [39]. (**n**) Ours.

**Figure 10 sensors-25-01889-f010:**
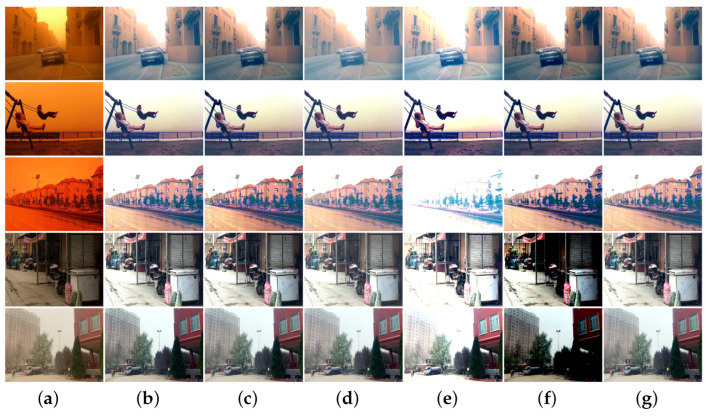
Ablation study of loss function. (**a**) Sand-dust images. (**b**) The proposed method. (**c**) Without atmospheric light loss $L_{A}$. (**d**) Without transmission loss $L_{T}$. (**e**) Without bright pixel saturation-penalized loss $L_{S}$. (**f**) Without dark pixel saturation-penalized loss $L_{S}$. (**g**) Without gradient loss $L_{V}$.

**Table 1 sensors-25-01889-t001:** Average results of 25 sand-dust images.

Method	e↑	σ↓	$\bar{r}↑$	NIQE ↓	DIIVINE ↑	NPQI ↓
GDCP [5]	0.9137	0.2291	1.6784	3.9694	29.5532	12.2856
RBCP [6]	1.4863	0.1112	1.5531	3.979	30.7981	13.3088
SCBCH [10]	0.6106	0.0389	1.6838	3.7222	28.3511	11.0898
NGT [9]	0.4109	0.0056	1.8223	3.7858	25.4106	11.1056
ROP [33]	0.8444	0.0132	1.7258	3.7487	24.7047	11.8019
BCGF [16]	1.8427	0.6997	3.2769	3.8843	25.5295	11.4881
TLS [18]	0.0956	0.0059	1.4965	3.9467	31.7507	12.2684
FBE [21]	1.9576	0.1413	2.8525	3.8325	24.1332	10.5324
TOENet [38]	0.3259	1.2011	1.6010	3.7264	26.1932	11.2599
HDCP [43]	0.9905	0.1396	3.8849	4.1574	25.7301	12.1145
DedustGAN [40]	1.2122	0.0128	1.9202	3.9902	13.9408	12.1147
SIENet [39]	1.2428	0.0094	1.8845	3.8055	20.9273	11.4862
Ours	1.5434	0.0043	2.1766	3.6477	27.5495	10.6046

↑ indicates the larger the better. ↓ means the smaller the better.

**Table 2 sensors-25-01889-t002:** Average result of 1070 various sand-dust images.

Method	e↑	σ↓	$\bar{r}↑$	NIQE ↓	DIIVINE ↑	NPQI ↓
GDCP [5]	-	0.0717	1.7391	4.5527	38.6125	14.7407
RBCP [6]	-	0.102	1.6658	4.6593	40.2739	15.6666
SCBCH [10]	-	0.0654	1.8983	4.3238	39.4204	13.5423
NGT [9]	-	0.00006	1.9001	4.3936	35.6688	13.5000
ROP [33]	-	0.0087	2.2026	4.3379	35.0144	12.9264
BCGF [16]	-	0.5430	3.5192	4.3246	34.0732	13.199
TLS [18]	-	0.0536	1.4965	4.5909	42.9845	15.0563
FBE [21]	-	0.1397	2.6998	4.3437	34.5536	12.9475
TOENet [38]	-	0.2507	1.7650	4.0669	37.1339	13.6199
HDCP [43]	-	0.0199	4.5529	4.5226	30.4738	13.7403
DedustGAN [40]	-	0.091	3.8539	4.0215	19.0229	13.5084
SIENet [39]	-	0.0975	2.1398	4.3661	30.2219	12.9198
Ours	-	0.0897	2.3312	3.959	38.1299	12.8000

↑ indicates the larger the better. ↓ means the smaller the better.

**Table 3 sensors-25-01889-t003:** Average results of 460 real sand-dust images captured in Xinjiang.

Method	e↑	σ↓	$\bar{r}↑$	NIQE ↓	DIIVINE ↑	NPQI ↓
GDCP [5]	0.6197	0.2597	1.6029	4.0974	29.2947	11.1746
RBCP [6]	0.3482	0.4284	1.5271	3.8703	32.9548	10.4301
SCBCH [10]	0.3637	0.135	1.3703	3.7694	32.9136	10.3171
NGT [9]	0.3506	0.0026	1.7707	3.7218	27.326	9.705
ROP [33]	0.4042	0.0179	1.6389	3.7486	28.0376	12.0185
BCGF [16]	0.6387	0.8428	2.3283	3.9563	32.9117	10.5054
TLS [18]	0.2137	0.5202	1.4552	4.0519	38.49	10.5123
FBE [21]	0.5723	0.1614	1.7351	3.7598	29.1293	10.0479
TOENet [38]	0.4318	1.0574	1.4812	3.7103	28.3351	12.0144
HDCP [43]	0.5127	0.1478	3.4789	4.3531	23.5872	11.4722
DedustGAN [40]	0.5383	0.0258	1.9408	3.7074	16.3792	11.4295
SIENet [39]	0.54.1	0.0054	1.6834	3.8879	24.0551	11.2554
Ours	0.5633	0.2225	1.5227	3.6809	29.5885	10.6025

↑ indicates the larger the better. ↓ means the smaller the better.

**Table 4 sensors-25-01889-t004:** Ablation results of 1530 real sand-dust images with different loss functions.

$L_{A}$	$L_{T}$	$L_{V}$	$L_{\mathrm{SMx}}$	$L_{\mathrm{SMn}}$	σ↓	$\bar{r}↑$	NIQE ↓	DIIVINE ↑	NPQI ↓
✗	✓	✓	✓	✓	0.1471	1.7718	5.4957	26.1135	19.4831
✓	✗	✓	✓	✓	0.0275	1.5897	5.5769	32.0269	16.9101
✓	✓	✗	✓	✓	0.1127	1.7621	5.3012	26.8182	19.7357
✓	✓	✓	✗	✓	0.0451	2.2618	7.1590	31.9026	24.9664
✓	✓	✓	✓	✗	12.9885	1.8439	6.2291	30.0502	20.4819
✓	✓	✓	✓	✓	0.1561	1.927	3.824	33.8592	11.7013

↑ indicates the larger the better. ↓ means the smaller the better.

## Data Availability

The data presented in this study are available on request from the corresponding author. The data are not publicly available because the data were collected by the authors themselves.

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
