# Peer review of "Zero-Shot Sand-Dust Image Restoration"

_sensors, 2025, doi:10.3390/s25061889_

Round 1

Reviewer 1 Report

Comments and Suggestions for Authors

The authors propose a zero-shot learning method based on an atmospheric scattering physics model for sand-dust image restoration. Both qualitative and quantitative experiments are done to demonstrate the superior performance of the proposed method. Below are the comments.

1. In Section 3, the explanation of some notations in equations is insufficient. For example, the meaning of “Ig(x)” in Eq. (1) should be clarified.

2. In Section 3.3, the description of Fig. 4 is inadequate. It is recommended to provide additional information, such as the input and output sizes of the network, a detailed explanation of the “FC64-3” structure, and the meaning of dashed lines. Additionally, it would be helpful to specify which parts of Fig. 4 correspond to the three networks, MT, MJ, and MA, respectively.

3. The performance improvement resulting from the proposed color balance algorithm is not clearly quantified. It is recommended to include ablation experiments to demonstrate the effectiveness of this algorithm.

4. In Table 1-Table 3, the DIIVINE metric for RBCP, SCBCH, and TLS consistently outperforms the proposed method. This discrepancy should be addressed and explained.

5. The reference format needs to be unified and standardized to comply with the required guidelines. For example, some journal or conference names are italicized and abbreviated, while others are not.

Author Response

Comments:

 In Section 3, the explanation of some notations in equations is insufficient. For example, the meaning of “Ig(x)” in Eq. (1) should be clarified.

Response:  

&nbspThank you very much for your comments. Considering the reviewer’s suggestion, we have modified the  explanation of Equation t. Modified content are marked in yellow on page 5, line 197-198.

Comments:

 In Section 3.3, the description of Fig. 4 is inadequate. It is recommended to provide additional information, such as the input and output sizes of the network, a detailed explanation of the “FC64-3” structure, and the meaning of dashed lines. Additionally, it would be helpful to specify which parts of Fig. 4 correspond to the three networks, MT, MJ, and MA, respectively..

Response:  

&nbspThank you very much for your comments, these comments are very helpful for revising and improving our paper. We have added the additional information. Modified content are marked in yellow on page 5 and 6, line 254-258. The input image size is explained on page 8, line 295-296.

Comments:

The performance improvement resulting from the proposed color balance algorithm is not clearly quantified. It is recommended to include ablation experiments to demonstrate the effectiveness of this algorithm.

Response:  

We sincerely appreciate the reviewer's valuable suggestion regarding the quantitative evaluation of our color balance algorithm. In the framework proposed in this article, if the sand dust images are not color corrected, the designed network will incorrectly estimate atmospheric light and transmission map, and cannot obtain accurate results. Therefore,  we did not conduct a quantitative evaluation of the color shift correction results in this article, and we will definitely fully consider the opinions raised by the reviewers in future research.

Comments:

In Table 1-Table 3, the DIIVINE metric for RBCP, SCBCH, and TLS consistently outperforms the proposed method. This discrepancy should be addressed and explained.

Response:  

Thank you very much for your comments. According to the reviewer’s suggestion, we have provided an explanation,  Modified content are marked in yellow on page 14, line 388-395.

Comments:

The reference format needs to be unified and standardized to comply with the required guidelines. For example, some journal or conference names are italicized and abbreviated, while others are not.

Response:  

Thank you very much for your comments. According to the reviewer’s suggestion, we have modified the references.

Reviewer 2 Report

Comments and Suggestions for Authors

In this paper authors try to address the sand-dust image restoration problem by introducing a novel zero-shot learning method based on the modelling of atmospheric scattering physics. 

Traditional methods and deep learning methods heavily rely on the prior estimation of the transmission and atmospheric light. Since real sand-dust images have high complexity, the prior assumption of typical methods, limits their robustness and generalisation.

The authors aim to cope with this problem, by using unsupervised and zero-shot learning approach. Moreover, the proposed method does not require prior knowledge or specific parameter tuning for the proposed model, which makes it robust.

The proposed framework consists of the three major modules, the colour balance algorithm, the model training and the restoration model.

The authors should add more details on the Ablation Study about the weights of loss function, providing the specific values of the weights used for the figures in Fig.10.

The complexity of the proposed and compared methods should also be included.

Experimental results are convincing. 

The conclusion section is consistent with the evidence presented in the previous sections.

Reviewer 3 Report

Comments and Suggestions for Authors

This work presents a new zero-shot learning method based on an atmospheric scattering physics model to restore sand-dust images. Particularly, a new joint-learning network structure model is proposed, and the network model is designed entirely based on the imaging principles of the physical model of atmospheric scattering. Experimental results demonstrated the effectiveness of the proposed framework. Overall, the paper is easy to follow and the performance seems to be promising. However, I have some concerns as follows and would like to raise my rating if the authors could properly address them.

- The details of the symbols used in Figure 1 are expected to be thoroughly described in its caption.

- Some related works regarding image restoration methods are missing. For example, "Denoising as adaptation: Noise-space domain adaptation for image restoration", "Test-Time Degradation Adaptation for Open-Set Image Restoration", "Domain Adaptation for Underwater Image Enhancement", etc. The authors are suggested to briefly review these works in the Introduction part to enrich the research scope of this work.

- Could the proposed framework be applied to more image restoration tasks such as underwater enhancement? I believe there exist some modeling overlaps among different tasks. Some insightful discussions would be beneficial.

- Since the proposed framework is primarily designed based on U-Net architecture, its unique contributions should be highlighted compared to the recent works that also used U-Net. For example, the related works used U-Net and proposed different modifications: "DR-GAN: Automatic radial distortion rectification using conditional GAN in real-time", "MOWA: Multiple-in-One Image Warping Model", "Blind Geometric Distortion Correction on Images Through Deep Learning", etc. These methods are expected to be discussed in the Related Work part to highlight the contribution of this work.

- Can the restored images of this work help the downstream vision tasks such as object detection and semantic segmentation? Since the visual enhancement quality may not align with the machine vision performance, I am wondering how the proposed low-level method benefits high-level vision models. Some qualitative comparison results would make the experiment more solid.

Comments on the Quality of English Language

The presentations look good to me.

Round 2

Reviewer 1 Report

Comments and Suggestions for Authors

1.It seems there are some annotations in the manuscript that do not comply with the required format. Please review the formatting of the manuscript again and remove them.
2.Please check the meaning of "FCM64-3" at line258
3.In all tables of the manuscript, to more intuitively demonstrate the superiority of the proposed method, the authors are advised to highlight the metrics where their approach shows advantages.

Reviewer 3 Report

Comments and Suggestions for Authors

Thanks for the authors' response. However, they only addressed some of my concerns. For the highly related works, I didn't see any changes in the main paper, such as the brief review and useful discussions of the suggested literature. The authors only replied to my concerns in the response letter, but all of them are expected to be reflected in the main paper. Thus, I cannot recommend accepting this work based on the current version.
